# Deep learning-based idiomatic expression recognition for the Amharic language

**Demeke Endalie**[1]*, **Getamesay Haile**[1], **Wondmagegn Taye**[2]

**1** Faculty of Computing and Informatics, Jimma Institute of Technology, Jimma, Ethiopia, **2** Faculty of Civil and Environmental Engineering, Jimma Institute of Technology, Jimma, Ethiopia

* demeke.endalie@ju.edu.et

**Data Availability Statement:** This research work's data set and source code are publicly available on GitHub. The link to access the data is: https://github.com/demekeendalie/Idiomatic-expression-.

## Abstract

Idiomatic expressions are built into all languages and are common in ordinary conversation. Idioms are difficult to understand because they cannot be deduced directly from the source word. Previous studies reported that idiomatic expression affects many Natural language processing tasks in the Amharic language. However, most natural language processing models used with the Amharic language, such as machine translation, semantic analysis, sentiment analysis, information retrieval, question answering, and next-word prediction, do not consider idiomatic expressions. As a result, in this paper, we proposed a convolutional neural network (CNN) with a FastText embedding model for detecting idioms in an Amharic text. We collected 1700 idiomatic and 1600 non-idiomatic expressions from Amharic books to test the proposed model's performance. The proposed model is then evaluated using this dataset. We employed an 80 by 10,10 splitting ratio to train, validate, and test the proposed idiomatic recognition model. The proposed model's learning accuracy across the training dataset is 98%, and the model achieves 80% accuracy on the testing dataset. We compared the proposed model to machine learning models like K-Nearest Neighbor (KNN), Support Vector Machine (SVM), and Random Forest classifiers. According to the experimental results, the proposed model produces promising results.

## 1. Introduction

In recent years, the development of deep learning in neural networks has improved performance in many natural language processing (NLP) tasks. In natural language processing, neural networks are used to develop machine translation, speech recognition, text generation, text mining, and named entity recognition.

An idiomatic expression is a phrase or expression whose meaning may be different from the combination of literal meanings of its composing words. The meaning of the idioms cannot be interpreted from the meaning of words that construct them directly [1]. Idiomatic expressions are one of the important parts of all-natural languages [2]. Amharic is one of the languages grouped under the Semitic language families [3]. The Amharic language has more than 4000 idiomatic expressions. The detection of this type of expression from Amharic text helps those individuals who are not familiar with the Language. For example, the expression

**Funding:** The author(s) received no specific funding for this work.

**Competing interests:** The authors have declared that no competing interests exist.

"ፊቱን ጣለዉ." (he drops his face) can be directly taken as the guy drops his face somewhere, but the actual meaning is "he becomes sad."

## 2. Related works

Idiomatic expression recognition from a given text plays an important role in implementing tasks such as machine translation, speech recognition, sentiment analysis, and dialog systems within the respective language. Idiom token classification involves determining if a phrase is literal or idiomatic [4]. Salton et al. [4] used Skip-Thought Vectors to create distributed representations with predictive features. Skip-thought vectors are generated using an encoder-decoder model. After receiving the training sentence, the encoder creates a vector. Encoders include uni-skip, bi-skip, and combine-skip, with uni-skip reading the text from beginning to end, bi-skip concatenating forward and backward results, and combined-skip concatenating vectors. Classifiers perform competitively, using only the target phrase as input, and are less dependent on discourse context. This approach can be used to train a competitive general idiom token classifier.

The study [5] uses a computational search approach to examine idiomatic language identification in non-native English writings. Idioms are often employed in English as a Foreign Language (EFL) essays, and a search method that considers their syntactic and lexical flexibility enhances recall by 30% and increases false positives.

Afsaneh et al. [6] used the connection between idiomaticity and (in)flexibility to create statistical measures to automatically distinguish idiomatic from literal verb plus noun combinations. VNICs differ in flexibility but contrast with compositional phrases, which are more lexically productive and have a wider range of syntactic forms. Lexical and syntactic flexibility can be used as partial indicators of semantic analyzability and idiomaticity.

An algorithm is proposed for the automatic classification of idiomatic and literal expressions [7]. It hypothesizes that high-ranking words in a text segment are less likely to be part of an idiomatic expression. The algorithm uses Latent Dirichlet Allocation (LDA) to extract topics from paragraphs containing idioms and literals. Idiomatic expressions are treated as semantic outliers, and outlier detection is used to distinguish idioms from literals using local semantic contexts.

In the study of [8], the authors proposed an idiomatic expression detection method based on the assumption that idioms and their literal counterparts do not occur in the same contexts. The inner product of context word vectors with the vector representing a target expression is computed first by their model. Because literal vectors predict local contexts well, their inner product with contexts should be greater than idiomatic ones. This distinguishes literals from idioms and, in word vector space, computes literal and idiomatic scatter (covariance) matrices from local contexts. Because the scatter matrices represent context distributions, they used the Frobenius norm to calculate their difference.

The work of [9] presents a generalized model for determining whether an idiom is used figuratively or literally based on the concept of semantic compatibility. They examine continuous bag-of-words (CBOW's) limitations regarding semantic compatibility measurement and propose a novel semantic compatibility model based on CBOW training for idiom usage recognition. Experiments on two benchmark idiom usage corpora reveal that the proposed generalized model outperforms state-of-the-art per-idiom models at the time.

In [10], a model for detecting idiomatic phrases in written text was proposed. This paper presents a binary classification approach for identifying idioms at the sentence level, offering insights into contexts and unique properties. The authors aim to improve detection rates using textual cohesion and compositionality measures. Textual cohesion refers to the grammatical

and semantic relationships between phrases or pieces of a text that contribute to unity and coherence. Textual compositionality measurements, on the other hand, assess the compositional aspects of a text. They used principal component analysis for idiom detection, linear discriminant analysis for discriminant subspace generation, and three nearest neighbor classifiers to obtain accuracy. They also analyzed the advantages and disadvantages of each technique, which are broader than previous idiom identification algorithms.

Idiomatic expression in language has a detrimental impact on improving language learning proficiency and NLP task performance [11, 12]. However, according to our best knowledge, no Amharic natural language processing model considers idiomatic expressions. This inspired us to create an Amharic idiomatic phrase identification system based on deep learning. This study focuses on constructing a CNN using the FastText model to detect the presence of idiomatic terms in an Amharic text. The overall contributions of the study are summarized as follows:

1. Prepare a general-purpose Amharic idiomatic expression dataset that can be used by other studies in the future.

2. Proposed a deep learning model incorporating CNN with FastText to recognize idioms from Amharic texts.

3. Evaluate the performance of the proposed recognition model with various evaluation metrics.

The remainder of the paper is structured as follows. Section 2 presents the state of art learning models. Section 3 presents the planned work's comprehensive methodology in detail. Section 4 defines the experimental results. In this section, we present the outcome and a discussion of it. Finally, section 5 is the conclusion.

## 3. Learning model

### 3.1. Convolutional neural network

We need a learning model to determine whether a particular phrase is idiomatic or not. A convolutional neural network is an advanced neural network model that discovers patterns and relationships between data items based on their relative positions [13]. CNN can automatically learn effective feature representation from massive text using a 1D structure (word order) in the convolutional layer. It captures local relationships among the neighbor words in terms of context windows, and by using pooling layers, it extracts global features. CNN is a neural network made up of several convolutional and pooling layers.

### 3.2. K-Nearest Neighbor classifier

K-Nearest-Neighbors is a basic yet effective non-parametric supervised classification technique. The KNN classifier is the most common pattern recognition classifier because of its effective performance, efficient outputs, and simplicity. It is frequently utilized in pattern recognition, machine learning, text classification, data mining, object identification, and various other domains [14]. The KNN method classifies by analogy, which means that it compares the unknown data point to the training data points to which it is comparable. The Euclidean distance is used to calculate similarity. The attribute values are adjusted to prevent bigger range characteristics from outweighing smaller range ones. In KNN classification, the unknown pattern is assigned the most predominant class amongst the classes of its nearest neighbors. In the event of a tie between two classes for the pattern, the class with the minimum average distance

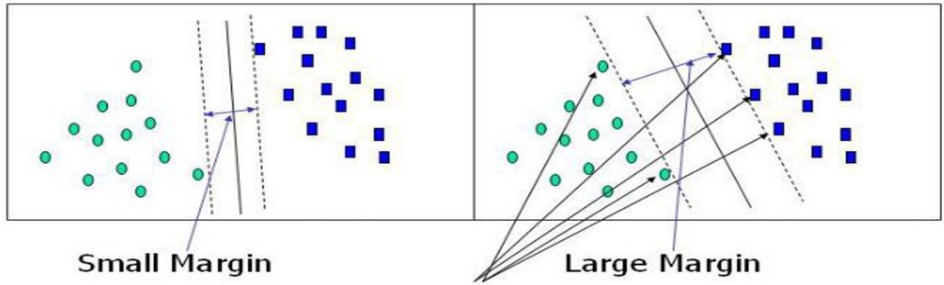

**Fig 1. Support Vector.**

to the unknown pattern is assigned. A global distance function based on individual attributes can be calculated by combining several local distance functions based on distance [15].

### 3.3. Support Vector Machine

Support Vector Machines and Kernel methods have found a natural and effective coexistence since their introduction in the early 90s. SVMs use kernels for learning linear predictors in high-dimensional feature spaces [16]. The objective of the SVM algorithm is to find a hyper-plane in N-dimensional space (N is the number of features) that distinctly classifies the data points. Hyperplanes are decision boundaries that help classify the data points. Data points on either side of the hyperplane can be attributed to different classes. Also, the dimension of the hyper-plane depends upon the number of features. If the number of input features is two, then the hyper-plane is just a line. If the number of input features is three, the hyper-plane becomes a two-dimensional plane. Fig 1 shows a sample decision boundary separation.

### 3.4. Random Forest classifiers

A random forest is a technique used in modeling predictions and behavior analysis and is built on decision trees. It contains many decision trees, each representing a distinct instance of the classification of data input into the random forest. The random forest technique considers the instances individually, taking the one with the majority of votes as the selected prediction [17]. A random forest generates a set of decision trees. To achieve diversity among basic decision trees, random forest chose the randomization approach, which works well with bagging or random subspace methods [18]. To generate each tree in the random forest, the following points should be considered: If the number of records in the training set is N, N records are randomly sampled but replaced by the original data. This is a bootstrap sample. This sample will be a training set for growing the tree. If there are M input variables, a number $m << M$ is selected, and at each node, m variables are randomly selected from M, and the best split over these m attributes is used to split the node. The value of m remains constant during forest growth. Each tree is cultivated to the best of its ability.

## 4. Materials and methods

FastText is an open-source text representation and categorization framework developed by Facebook's AI Research (FAIR) team. Using the FastText word vector representation, we generate a vector for each word in the corpus that can be directly fed into any learning algorithm. The goal of this study is to develop a deep learning model that uses FastText to detect the presence of idiomatic words in an Amharic text. Fig 2 below depicts the proposed idiomatic expression recognition architecture for the Amharic language. Pre-processing, word

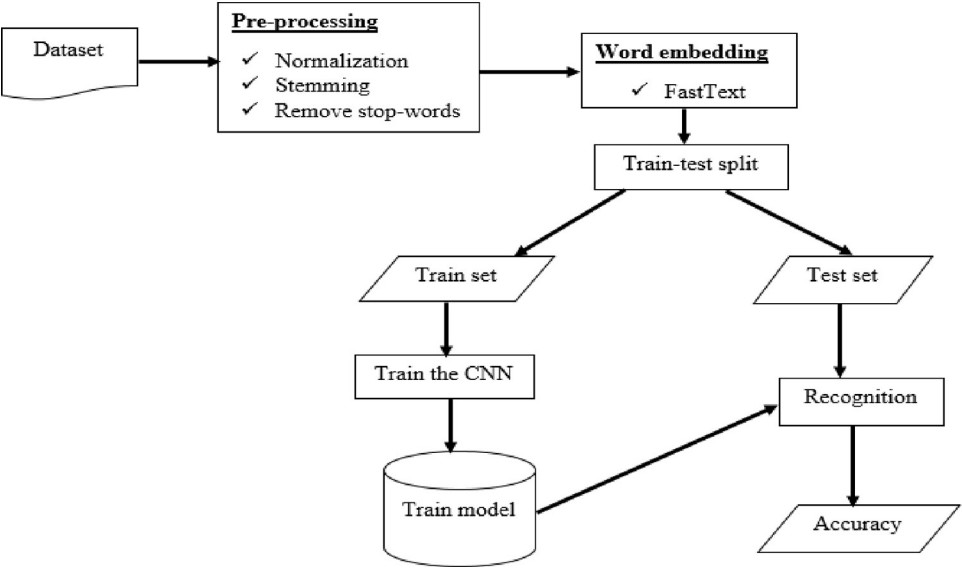

**Fig 2. The architecture of the proposed idiomatic expression recognition system.**

embedding, and learning modules are all components of the proposed automatic idiomatic expression identification system. The tasks in the proposed model range from data gathering to evaluation. This means the proposed model contains tasks from data collection up to evaluation.

## 4.1. Dataset

This study concentrated on detecting Amharic idiom types. This means detecting whether a phrase is usually an idiom or usually literal. The dataset for the study is gathered and processed with this major goal in consideration. The dataset utilized in this study was gathered from two Amharic books "የአማርኛ ፈሊጦች" (idiomatic expressions in Amharic), and "ፍቅር እስከ መቃብር" (love up to the grave) [3, 19]. The book writers already annotate the data as idiomatic or literal. The idiomatic terms used in this study were hand-picked and cross-checked from published books.

Most idiomatic expressions in our source books have 2 to 4 tokens, so the dataset contains only 2 to 4-length idiomatic expressions. There are more than four thousand idioms in the Amharic language. In this study, we used 3300 isolated Amharic idiomatic and non-idiomatic expressions to train and evaluate the proposed model. Out of 3300 Amharic phrases, 1700 were idiomatic expressions, and 1600 were phrases containing terms found in idiomatic expressions but not utilized as an idiom. The idiomatic phrases were compiled from the afore-mentioned publications and are easily readable in the books themselves. The 1600 non-idio-matic phrases were collected from those publications.

Let us use an example to clarify the distinction between idiomatic and non-idiomatic phrases in the dataset." እግረ ደረቅ ነው" (igire derek'i newi), is an idiomatic phrase which has the meaning "the feet are dry" when we interpret the meaning of each word, but the actual meaning of the phrase is "Unlucky."" እግረ አባጣ ነው" (igirē ābat'a newi) is a non-idiomatic phrase which means "he has elephantiasis." This is the only meaning that can be derived from the literal meanings of each word in it. Despite the fact that both phrases utilize the same word, "እግረ," which means leg, the first phrase does not refer to its real meaning, leg, whereas

**Table 1. The dataset's idiomatic expression length distribution.**

| Token count in the idiomatic expression | The total number of idiomatic expressions of the given length |
|---|---|
| Idiomatic with two tokens (words) | 1620 |
| Idiomatic with three tokens (words) | 70 |
| Idiomatic with four tokens (words) | 10 |

The dataset's non-idiomatic clauses have a length of two up to four tokens. After collecting the data from books, we apply the following preprocessing modules to clean up the data and make the learning phase as easy as possible.

the second phrase does. We use text processing activities such as stop word removal to remove terms such as "ነው" (newi) that have no bearing on the meaning of the statement.

Idiomatic phrases were labeled zero, and non-idiomatic phrases were labeled one in the dataset. Table 1 shows the distribution of the length of idiomatic statements in the collection.

**i. Normalization.** The Amharic writing system has different letters ("") that can be read with the same pronunciation, but there are no rules to distinguish their meanings. As a result, in Amharic, these letters may represent the same concept or name of an object. For instance, the word "power" can be written as ኃይል፣ ሀይል፣ ሐይል፣ ሓይል፣ ጎይል፣ኃይል. These six terms do not have meaning differences but use different letters with the same phonetics. The normalization of Amharic characters is similar to that of other Semitic language families, such as Arabic and Hebrew [20]. This increases the number of features extracted for processing or analysis. To avoid this inconsistency, normalize those characters or letters with the same phonetics to one common canonical form. To overcome this redundancy, we normalize those characters with the same pronunciation to one canonical letter used in this study, as shown in Table 2 below [21]. Normalization aims to reduce the number of distinct features in the gathered dataset.

**ii. Stemming.** Stemming is the process of reducing inflected words to their stem, base, or root form. Amharic is one of the morphological-rich Semitic languages [22]. Different terms can exist with the same stem, and this helps reduce the size of feature space for processing. In this study, we used the HornMorpho stemmer developed by Michel Gasser [23]. HornMorpho is a Python library developed to analyze three Ethiopian languages: Amharic, Afan Oromo, and Tigrigna.

**iii. Remove stop words.** In Amharic, the common words, e.g., "ሁሉ፣ እስከ፣ ነው፣ሆነ ", and others that scoreless weightage in the text processing tasks is called stop words. In this study, stop words are terms that do not contribute to the semantics of the given idiomatic phrase but are used to fill the grammatical structure of the idiomatic statement. Stop words are eliminated to save computational time wasted in processing them. Amharic does not have a well-prepared list of stop words. However, we remove stop words prepared by [21]. In addition, to stop word removal, we also replace numbers with their name in alphabetic characters ("ፊደል"). To obtain the feature vectors of the numbers' names, we use the FastText embedding and substitute

**Table 2. Normalization of characters having the same pronunciations.**

| Canonical character | Characters with the same pronunciation as the canonical character |
|---|---|
| U(hā) | ሃ፣ኃ፣ኀ፣ሐ፣ሓ(hā) |
| ሰ (še) | ሠ (še) |
| አ(ā) | እ፣ዐ፣ዓ(ā) |
| ጸ(ts'e) | ፀ(ts'e) |
| ዉ(wu) | ዑ(wu) |

numbers with their alphabetic characters. In the pre-trained FastText-based word embedding, only Amharic alphabetic characters are employed. As a result, for this research, numbers must be translated to their alphabetic names. For example, in "2 ኣይን" (two eyes), 2 can be changed to two ("ሁለት") and produce "ሁለት ኣይን". This replacement is done by keeping a map of the key-value relation between digits and an alphabetic description of each digit.

## 4.2. Text representation

Encoding is extensively required to pass texts as input to different machine learning and deep learning models [24]. One of the text encoding algorithms that changes a given text into a vector is the word2vec algorithm. It is a set of neural network models used to represent a word in a vector space. Those words which have similarities in their context are clustered together, and those that do not have any contextual meaning similarity appear sparsely on the vector space. However, word2vec fails to generate the vector of words that are not in the training vocabulary.

FastText is one of the state-of-the-art word embedding models developed by Facebook. For 157 languages, Facebook develops pre-trained FastText embedding models. One of the languages with a trained FastText word embedding model is Amharic. FastText embedding's strength is that it can create a vector for a given term even if it is not in the training vocabulary. This is resolved by considering the character-level n-gram of a given term. Each idiomatic expression's embedding is constructed using pre-trained FastText embedding. The embedding is then saved as a matrix in which the rows reflect the number of idiomatic expressions in the dataset, and the columns represent the embedding dimension used by the embedding technique. The embedding is built before splitting the dataset into training and testing sets.

## 5. Results and discussions

All experiments are carried out in a Windows 10 environment on a machine equipped with a Core i7 processor and 16 GB of RAM. The accuracy, precision, recall, and f1-score are used to assess the performance of the models used in this study. The formulas used to calculate them are shown in Table 3 below.

Where Tp denotes true positive, Tn denotes true negative, Fp denotes false positive, and Fn denotes false negative. The experimental configurations required to construct the proposed Amharic idiomatic expression recognition system are determined during experimentation using grid search-based adjustment.

## 5.1. Training and validating the model

We have divided the data to train and validate its performance with a training test split ratio of 80%, 10%, and 10% for training, validating, and testing the proposed model, respectively. We did not employ k-fold cross-validation to assess the performance of the proposed model

**Table 3. Performance evaluation metrics.**

| Evaluation metric | Formula |
|---|---|
| Accuracy | $accuracy = \frac{Tp+Tn}{Tp+TN+Fp+Fn}$ |
| Precision | $precision = \frac{Tp}{Tp+Fp}$ |
| Recall | $recall = \frac{Tp}{Tp+Fn}$ |
| F1-score | $f1-score = \frac{2(recall*Precision)}{recall+precision}$ |

**Table 4. Hyperparameter values of the CNN model.**

| Hyperparameters | Values |
|---|---|
| Embedded dimension | 300 |
| Number of filters | 265 |
| Batch size | 16 |
| Dropout | 0.5 |
| Activation | Sigmoid |
| Optimization | Adam |
| Epoch | 100 |
| Loss | Binary cross entropy |

because the dataset is small, and a portion of it is used for model evaluation, reducing the amount of data available for training the model indirectly [25]. We used unique idiomatic and non-idiomatic terms (those not included in the training set) to validate and evaluate the proposed idiomatic recognition model. We tune the hyperparameters using a grid search strategy to train the proposed CNN model. The training, validating, and testing sets are separated after producing the dataset's word2vec of each idiomatic expression. There are 1360, 168, and 172 instances of idiomatic expressions used in training, validating, and testing, respectively. The non-idiomatic expression training, validation, and testing examples are 1280, 160, and 160, respectively. The value of the hyperparameters used in this study is shown in Table 4 below.

To feed the training, validating, and testing data to the CNN model, we first create three matrices for the training, validation, and testing sets, each with dimensions of (number of training samples,300), (number of validating samples,300), and (number of testing samples,300). The number 300 is chosen for the matrix because the dimension of the matrices we build must be equal to the embedding matrix value of the CNN, as mentioned in Table 4 above. Next, using looping, fill the above matrix with the vector values of the FastText embedding model's idiomatic and non-idiomatic words. Following that, we train the CNN model by supplying the training data matrix and training data label and validate its performance by passing the validation set's matrices with the hyperparameter values shown in Table 4.

The model's training accuracy and loss are then displayed in Figs 3 and 4 below after the model has been trained using the above-mentioned parameters and training dataset. Since the training accuracy grows as the number of epochs increases, the model learns from the data well. In addition, as the number of epochs rises, the training loss declines. This shows that the model picks up on idiomatic expression features from the training set.

## 5.2. Testing the model

With the testing dataset and the evaluation metrics listed in Table 3 above, we assess the effectiveness of the proposed Amharic idiomatic expression recognition model. Fig 5 below shows the experimental results of how well the proposed scheme performed in terms of accuracy, precision, recall, and f1-measure.

As shown in Fig 5 above, the proposed Amharic idiomatic expression recognition system, which makes use of CNN with FastText word embedding, achieved results with accuracy, precision, recall, and f1-score of 80%, 70%, 77.78%, and 73.68%, respectively. The 80% means that the proposed idiomatic detection approach correctly detected 266 idiomatic and non-idiomatic phrases out of 332 cases in the testing dataset. The proposed idiomatic recognition model fails to recognize the remaining 66 testing instances. The results show that deep learning with FastText embedding can detect Amharic idiomatic expressions in a text with better

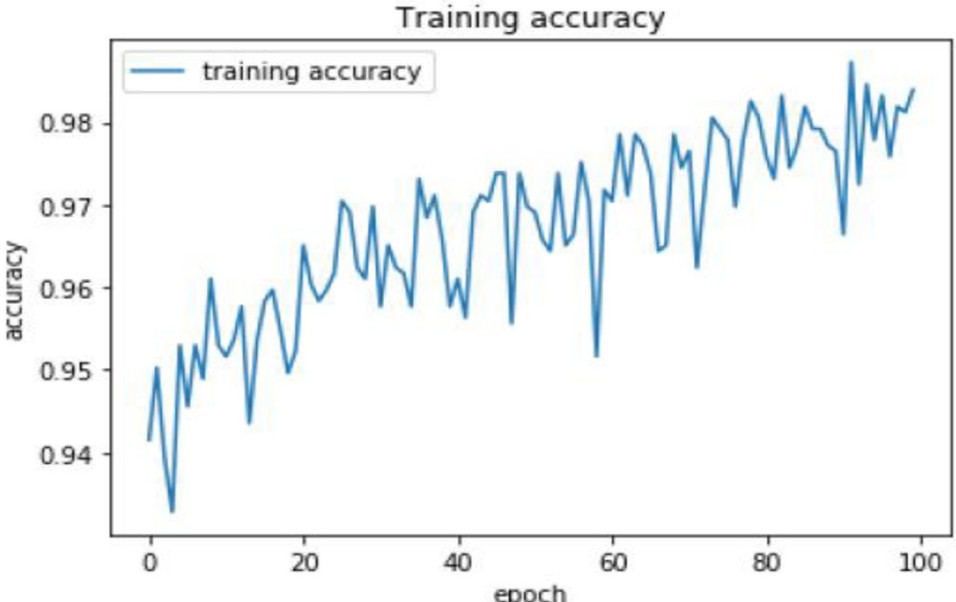

**Fig 3. Training accuracy of the proposed model.**

results across multiple quality criteria. This means that researchers or application developers will utilize deep learning-based idiomatic expression identification models to improve the performance of their models or applications.

## 5.3. Comparison of the performance of the model with other models

We must consider two factors to justify a model working effectively [26]. These factors are 1) by examining the model's numerical output and 2) by contrasting its performance with that of

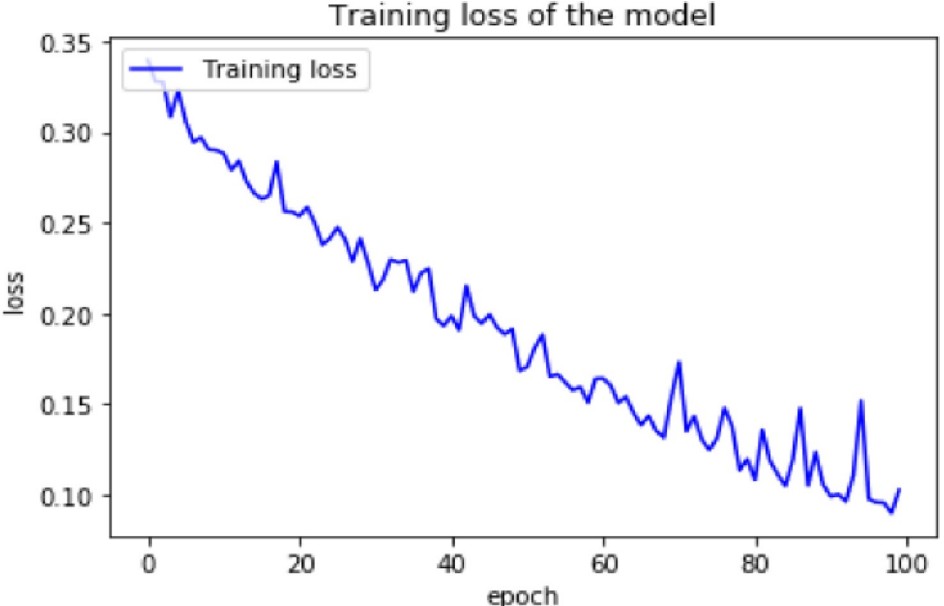

**Fig 4. Training loss of the proposed model.**

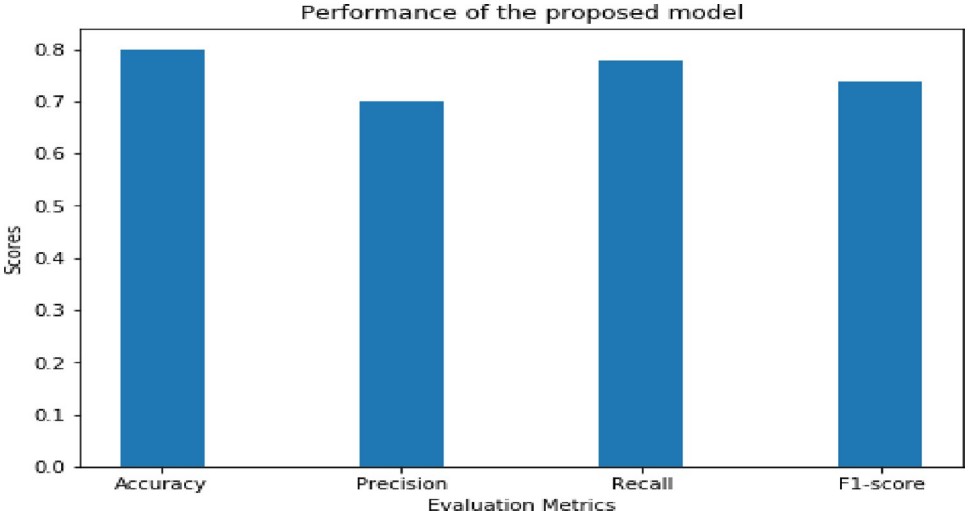

**Fig 5. Evaluation of the performance of the proposed model with accuracy, precision, recall, and f1-score.**

other models applied to the same dataset by other studies. As a result, we contrasted the new model's performance with some of the machine learning models employed in earlier studies [27]. We compare the proposed model against KNN, SVM, and Random Forest classifiers.

The vector generated by the pre-trained FastText retained the meanings of each idiomatic and non-idiomatic phrase. This vector for the training and testing set, along with their label, is passed to the machine learning algorithms employed in this study, as explained in section 4.1. The hyperparameter values of these learning models were adjusted using a grid search-based tuning approach. Table 5 shows the optimal hyperparameter values found using this grid-searching approach.

Table 6 below shows the detection accuracy these learning models using the hyperparameter value from Table 5. In addition, Table 6 is used to compare the detection accuracy of different learning models against the proposed deep learning with FastText-based word embedding for Amharic idiom detection. The number 72%,68%, and 76% indicates the percentage of the testing set correctly detected by Random Forest, KNN, and SVM models, respectively. To determine whether or not the aforementioned machine learning algorithms appropriately label a given phrase as idiomatic or non-idiomatic. To begin, we must use FastText to create a vector of the phrase with the same vector dimension as the one used for training.

All the above results shown in Table 6 above are produced with the same dataset and with the same word embedding model, which is FastText. The results show that the proposed algorithm enhanced detection accuracy by 8%, 12%, and 4%, respectively, compared to Random Forest, KNN, and SVM. This is since (1) employing FastText to generate word vectors can yield important features, and (2) processed text features using CNN can better represent high-level characteristics in the given idiomatic and non-idiomatic phrases.

**Table 5. The optimum values of hyperparameters for KNN, SVM, and Random Forest learning models.**

| Learning algorithms | The optimal values of hyperparameter values | | | |
|---|---|---|---|---|
| KNN | n_neighbors = 2 | weights = 'uniform' | The default for other parameters | |
| SVM | Kernel = 'rbf' | C = 1.0 | Gamma = 1 | The default for other parameters |
| Random Forest classifier | n-estimators = 300 | Class_weight = 'none' | The default for other parameters | |

**Table 6. Comparison of the proposed model with SVM, KNN, Random Forest.**

| Models | Accuracy |
| --- | --- |
| Random Forest | 72% |
| KNN | 68% |
| SVM | 76% |
| Proposed model | 80% |

In addition to this, we compared the performance of the proposed idiomatic recognition model (CNN with FastText embedding) with other word embedding models like Term Frequency-Inverse Document Frequency (TF-IDF) and one-hot encoding vectors. IDF = Log [(# Number of documents) / (Number of documents containing the word)] and TF = (Number of repetitions of a word in a document) / (# of words in a document). The TF-IDF is calculated for each of the 3300 Amharic phrases used in this study. To produce our dataset's TF-IDF matrix, we first create a list of unique word lists and then compute the TF-IDF of each word in each phrase. The generated matrix has a dimension equal to the entire number of phrases in the dataset multiplied by the total number of unique words. We append the label of each phrase in the matrix to the end of each row. Then, we divide this matrix into training and testing and train each of the models used in this study. The result is depicted as shown in Table 7 below.

According to the results in Table 7 above, CNN with FastText is more effective at identifying idioms in Amharic language. The FastText embedding preserves the contextual meaning of every phrase in the dataset. Therefore, it outperforms both TF-IDF and one-hot encoding word embedding approaches. This is because the features of idiomatic expressions in the Amharic language can be gained better with the help of FastText's embedding [28]. By combining the benefits of CNN with the pre-trained FastText embedding model, the proposed approach detected Amharic idioms with higher accuracy than other machine learning models and words' vector representation. Even though the proposed model performed better in detecting idioms, it has to be supplemented with more idiomatic expressions annotated by Amharic linguistic specialists to improve its performance further. The enlarged model can then be used in Amharic machine translation, question answering, and sentiment analysis models or applications.

## 6. Conclusion

Different NLP models are now being developed for the Amharic language without considering idiomatic expressions. Models that do not take idiomatic recognition into account may produce incorrect results since the actual meaning of the expression differs from the meaning of each word that makes up the expression. Idioms are one of the most fascinating and difficult aspects of Amharic vocabulary. Machine learning algorithms do not process text as input, so they require encoding of texts into another format. We produced a vector of each word used in this study using pre-trained FastText word embedding as part of this encoding. The

**Table 7. Comparison of different words' vector representation.**

| Model | Word Embedding | Recognition accuracy |
| --- | --- | --- |
| CNN | FastText | 80% |
| | TF-IDF | 74% |
| | One-hot encoding | 71.3% |

experimental findings show that compared to models utilized in this study, the proposed CNN with the FastText embedding model is more effective at detecting Amharic idioms. The proposed approach can, therefore, be applied to natural language processing tasks requiring the detection of idiomatic expressions, such as machine translation, sentiment analysis, and question-answering systems. Potentially, the model's performance could be improved by training on more data. In the future, other datasets from Amharic holy books, such as the Amharic Bible, will be included in the aforementioned model to improve its performance. Furthermore, we propose to use this model as a component in Amharic machine translation.

## Supporting information

**S1 Appendix. "Amharic stop words eliminated in this study".** "The English translation of the stop words eliminated in this study".
(PDF)

## Acknowledgments

The authors would like to thank the Jimma Institute of Technology for supporting them through different resources. The authors would like to thank Jimma University for its support during the research work.

## Author Contributions

**Conceptualization:** Demeke Endalie.

**Data curation:** Demeke Endalie, Wondmagegn Taye.

**Formal analysis:** Demeke Endalie.

**Funding acquisition:** Demeke Endalie.

**Investigation:** Demeke Endalie.

**Methodology:** Demeke Endalie.

**Resources:** Demeke Endalie.

**Software:** Demeke Endalie.

**Supervision:** Getamesay Haile, Wondmagegn Taye.

**Validation:** Demeke Endalie.

**Visualization:** Demeke Endalie.

**Writing – original draft:** Demeke Endalie, Wondmagegn Taye.

**Writing – review & editing:** Demeke Endalie, Getamesay Haile, Wondmagegn Taye.

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
