## [Decision Letter · Decision Letter 0]

7 Aug 2023

PONE-D-23-14519Deep Learning-Based Idiomatic Expression Recognition for the Amharic LanguagePLOS ONE

Dear Dr. Endalie,

Thank you for submitting your manuscript to PLOS ONE. After careful consideration, we feel that it has merit but does not fully meet PLOS ONE’s publication criteria as it currently stands. Therefore, we invite you to submit a revised version of the manuscript that addresses the points raised during the review process.

We look forward to receiving your revised manuscript.

Kind regards,

Michael Flor

Academic Editor

PLOS ONE

Journal Requirements:

Additional Editor Comments:

Please pay attention to the reviewer comments.

The paper certainly need revision along the lines outline in the review.

In addition, consider the following points:

“There are more than four thousand idioms in the Amharic writing system.”

May be “in the Amharic language” ?

Normalization of orthography needs to be better explained. Is the substitutability purely phonetic? Is it similar to what can be found in Arabic or Hebrew?

“we eradicate stop words” - 'eradicate' is not a suitable word here, use 'eliminate' or 'remove'.

Please explain why do you need to remove stop words for idiom recognition? Why do you need to replace numbers – this should be explained and motivated.

“Algorithm 1” is not needed. You can just say you load vectors for each word from the database.

But what does it mean”we created a vector of both idiomatic and non-idiomatic (literal) words” ?

Is that before training? Or is that a description of training? Please elaborate this statement.

For section 4: how are the idiomatic and non-idiomatic sentences distributed among the train, validateand test sets? Is there any reason why you did not use cross-fold validation?

What features were used for the KNN, RandomForest and SVM classifier? Please explain the components of those models for you data? For example – did you use KNN over averaged embeddings?

Some suggested references:

https://aclanthology.org/P16-1019/

https://aclanthology.org/W18-0905/

https://aclanthology.org/E09-1086/

Reviewers' comments:

Reviewer's Responses to Questions

**Comments to the Author**

1. Is the manuscript technically sound, and do the data support the conclusions?

Reviewer #1: Partly

2. Has the statistical analysis been performed appropriately and rigorously? 

Reviewer #1: N/A

3. Have the authors made all data underlying the findings in their manuscript fully available?

Reviewer #1: No

4. Is the manuscript presented in an intelligible fashion and written in standard English?

Reviewer #1: No

5. Review Comments to the Author

Reviewer #1: The authors claim in the abstract that idioms are used to conceal information. References to support this claim?

"With testing and training datasets, the proposed model achieves an accuracy of 80% and 98%, respectively."" What does it mean? Do you evaluate on the training data?'

ፊቱን ጣለዉ -- I'd suggest transliterating examples.

You probably don't mean that the total number of idioms in all Semitic languages is 4000, but it reads this way in "Amharic is one of the languages grouped under the Semitic language families that have more than 4000 idiomatic expressions [3]."

Many typos/sloppy writing, e.g., The Authors of [7] -- lower case?

Section 2 -- pretty standard. I don't know if it's important to devote the whole page to these algorithms or if it's enough just to list them with references.

While the preprocessing step is described in detail, many things remain unclear - why is it important to normalize the orthographic representation of words? How does phonetic ambiguity contribute to idiom detection?

It would be nice to understand what stop words were removed (maybe add an appendix). Also, why do you normalize the numerals? Why is it important?

section 3.3 -- the evaluation metrics are pretty standard as well. It's not a criticism, but I"m not sure if the authors need a separate section for it.

My main criticism of this paper is that the paper doesn't tell a story. The paper briefly mentioned the dataset (but doesn't really describe it), the representation, experiments, evaluation, numbers. So, what does it all mean? How hard is the task? What kind of idioms did you deal with? Explain your data collection and annotation. How did you obtain your dataset? Who annotated the data? What's the inter-rater agreement? Are there expressions that can appear idiomatically and literally depending on the context? I.e., do you have the same PIEs (potentially idiomatic expressions) that appear in both the idiomatic and non-idiomatic training/test set?

The experiments are pretty standard. The most interesting part here is the Amharic language and the potential to explore the specific challenges of Amharic, and why the CNN model with FastText representations outperforms the rest. How do you compare with the other approaches to idiom detection? None of the previous approaches is implemented and applied to the Amharic data for comparison. It would be really interesting to see how well they perform on this language. None of it is explored in the paper.

I would start with showing idiomatic expressions in Amharic in context and explain why they present a challenge for the current NLP models. What contribution does this paper make?

A lot of missing references. To mention just a few: Sporleder et al., Flor et al., Liu et al., Salton, etc. etc.

6. PLOS authors have the option to publish the peer review history of their article (what does this mean?). If published, this will include your full peer review and any attached files.

Reviewer #1: No

---

## [Author Response · Author response to Decision Letter 0]

19 Aug 2023

address all comments and suggestion given.

---

## [Decision Letter · Decision Letter 1]

4 Oct 2023

PONE-D-23-14519R1Deep Learning-Based Idiomatic Expression Recognition for the Amharic LanguagePLOS ONE

Dear Dr. Endalie,

Thank you for submitting your manuscript to PLOS ONE. After careful consideration, we feel that it has merit but does not fully meet PLOS ONE’s publication criteria as it currently stands. Therefore, we invite you to submit a revised version of the manuscript that addresses the points raised during the review process.

Please see editor comments below

We look forward to receiving your revised manuscript.

Kind regards,

Michael Flor

Academic Editor

PLOS ONE

Journal Requirements:

Additional Editor Comments:

Here are editor comments for the revised manuscript.

1.

Page 1:

"we proposed a conventional neural network (CNN)"

You probably mean 'convolutional', not 'conventional'.

2.

Page 1:

"Idiomatic expression is a collection of words that have a different meaning"

Idiomatic expression is not just a 'collection' of words.

Consider maybe this formulation:

"Idiomatic expression is a phrase/expression whose meaning may be different from the combination of literal meanings of its composing words. "

3.

Paragraph that has the sentence

"Idiom token classification involves determining if a phrase is literal or idiomatic [4]."

and the paragraph starting with

"The paper by [6] presents the use of Skip-Thought Vectors".

Those two paragraphs use two different references [4] and [6] but essentially refer to the same paper. In fact, reference [4] is correct, but [6] is wrong. Please rewrite the paragraph and eliminate reference [6]. Also, do not presume that readers know what Skip-Thought Vectors are, you need to provide an explanation of that term.

4.

Regarding the paragraph about reference [9], including the sentence:

"Idiom detection as lexical outliers does not make use of class label information."

Your description of that work does not make it clear what class labels might be involved and why they are relevant or not. Please make your description clearer.

5.

Paragraph starting with the sentence:

"Idiomatic expression in language has a detrimental impact on NLP task performance [10]."

The problem is that reference [10] discusses the importance of idioms for English language learners (humans), and not for NLP systems. Please revise your paragraph.

6.

Section 2.1,

This section talks about CNN, but has reference [11] (García-Laencina et al.), which is a reference to a paper about KNN. Please adjust your references.

7.

Section 2.4 Random Forest Classifiers.

This section includes the sentence "To achieve diversity among basic decision trees,

random forest chose the randomization approach, which works well with bagging or random

subspace methods [15]".

However, reference [15] (Patodkar and Sheikh) is not about random forest classifiers, rather it is about Twitter sentiment classification. Please refine your references.

8.

Section 3. Materials and Methods

This section begins with the sentence

"This study focuses on developing a deep learning model using FastText..."

It would be nice to tell the readers t this point what FastText is. Although that is explained later in the manuscript, it would be good to provide a hint at this point in the narrative. Maybe say "...using FastText embeddings...".

9.

Section 3.1. Dataset.

The description of the dataset is unclear.

The sentence "We received 1700 idiomatic words..." is strange, as you are handling expressions – so maybe you meant "1700 idiomatic expressions".

Next, the manuscript says "we also collect 1600 phrases that are not classified as idiomatic expressions".

Here is the crucial question: what is the nature of those 1600 phrases?

Are those phrases that could be idiomatic (in some other context), do those phrases overlap with the set of 1700 idiomatic expressions? (one would expect so!)

or are the 1600 expressions just some literal expressions that are unrelated to the idiomatic subset ?

It is very important to clarify this point. If the expressions in the set of 1700 and those in the set of 1600 are not the same, then we are not talking about idiom token classification.

See for example Adewumi et al. (2021), https://aclanthology.org/2022.lrec-1.72/

10.

Another clarification point about the dataset: what about the contexts of those collected expressions? Did those expressions appear in sentences?

It is not clear whether you classified just isolated expressions or expressions in context (in sentences).

This must be clarified and explained explicitly, in detail.

You might want to provide some examples from your sets of idiomatic and non-idiomatic cases (with English translations).

If you were classifying expressions in sentences, you need to provide some statistics – e.g. what was the average word-count per sentence in each of the subsets (idiomatic and non-idiomatic).

Please also indicate the average word-counts per subset after the cleaning and normalization procedures.

11.

Regarding section 3.2.

It is not clear how are you using the word embeddings. You get an embedding for each word – and then what? You need to describe how the different algorithms use the embeddings! You cannot just say 'embeddings' and then proceed to results. Describe how are the features and/or embeddings integrated in your CNN, KNN, SVM and RF classifiers.

This is is the most substantial comment for this manuscript.

12.

Regarding TF-IDF, please describe which corpus was used for obtaining the TF and IDF values.

13.

Just after Table 7, there is a sentence "The suggested model generally detected Amharic idioms with acceptable accuracy..." Whether accuracy of 80% is acceptable or not, is debatable. Please avoid self-congratulatory remarks.

14.

Section 5 "Conclusion" includes a sentence

"This misleads models since..." That is a very awkward phrasing in English. Models are not 'misled'. Models can be inaccurate or produce inaccurate results.

Regarding bibliographic references:

15.

"[2] Oktay Yağiz, "anguage, Culture, Idioms, and Their Relationship with the Foreign Language,"...

"L" is missing before "anguage".

16.

"[23] Ton Van der Valk, Jan H. Van Driel, Wobbe De Vos, "Common Characteristics of Models in Present-day Scientific Practice," Research in Science Education, vol. 37, no. 4, pp. 469-488, 2020."

The year of publication is 2007, not 2020.

Reviewers' comments:

Reviewer's Responses to Questions

**Comments to the Author**

1. If the authors have adequately addressed your comments raised in a previous round of review and you feel that this manuscript is now acceptable for publication, you may indicate that here to bypass the “Comments to the Author” section, enter your conflict of interest statement in the “Confidential to Editor” section, and submit your "Accept" recommendation.

Reviewer #1: All comments have been addressed

2. Is the manuscript technically sound, and do the data support the conclusions?

Reviewer #1: Yes

3. Has the statistical analysis been performed appropriately and rigorously? 

Reviewer #1: Yes

4. Have the authors made all data underlying the findings in their manuscript fully available?

Reviewer #1: Yes

5. Is the manuscript presented in an intelligible fashion and written in standard English?

Reviewer #1: Yes

6. Review Comments to the Author

Reviewer #1: My comments have been addressed. The rest will be addressed in future studies. I found the approach pretty standard and was looking for interesting insights given that it dealt with a lesser studied language.

7. PLOS authors have the option to publish the peer review history of their article (what does this mean?). If published, this will include your full peer review and any attached files.

Reviewer #1: No

---

## [Author Response · Author response to Decision Letter 1]

11 Oct 2023

We address all questions and suggestions as much as we can!

---

## [Editor Report · Decision Letter 2]

18 Oct 2023

PONE-D-23-14519R2Deep Learning-Based Idiomatic Expression Recognition for the Amharic LanguagePLOS ONE

Dear Dr. Endalie,

Thank you for submitting your manuscript to PLOS ONE. After careful consideration, we feel that it has merit but does not fully meet PLOS ONE’s publication criteria as it currently stands. Therefore, we invite you to submit a revised version of the manuscript that addresses the points raised during the review process.

We look forward to receiving your revised manuscript.

Kind regards,

Michael Flor

Academic Editor

PLOS ONE

Journal Requirements:

Additional Editor Comments:

After the second revision, the manuscript is now much better and clearer.

There are still a few points that need additional revision.

1.

In the abstract:

“We collected 1700 idiomatic

and 1600 non-idiomatic clause datasets from Amharic books”

Please delete the words 'clause datasets' and replace them with 'expressions'

2.

Page 2:

“The Amharic language contains more than 4000 idiomatic clauses (expressions).”

Please rephrase it as

“The Amharic language has more than 4000 idiomatic expressions.”

3.

Page 2.

Take the sentences

“Amharic is one of the languages grouped under the Semitic language families

[3]. The Amharic language has more than 4000 idiomatic expressions.”

and move them just after “all-natural languages [2].”

4.

Page 2.

Before the sentence “Idiomatic expression recognition from a given text plays....”,

please put a heading “Related work”.

5.

Concerning the notion of 'idiom token' classification (as mentioned in the section about related work).

After checking out the train- and test- files that are available on your github site,

it seems obvious that your research was focused on classification of phrase types rather than tokens.

This means detecting whether a phrase is usually an idiom or usually literal. This is a viable research direction, and is OK, but this needs to be very explicitly stated in the manuscript (in the “Dataset” section).

This contrasts with idiom token classification, which tries to decide whether a given phrase in a given context (sentence) is used idiomatically or literally. All the papers cited in your 'related work' section are focused on 'idiom token' detection. You should also cite a work that has focused on 'idiom type' detection; specifically I suggest this one: https://aclanthology.org/E06-1043/

The distinction between idiom type and idiom token classification must be made very clear in the manuscript.

6.

Page 2.

Instead of “This paper uses Skip-Thought Vectors...”,

please write “Salton et al. [4] used Skip-Thought Vectors...”

7.

Page 3.

“However, according to the researchers' understanding, no Amharic...”

please rephrase as

“However, according to our best knowledge, no Amharic...”

8.

Section 3.2

“Most idiomatic expressions in books are 2 to 4 in a number of tokens exist,”

please rephrase it as

“Most idiomatic expressions in our source books have 2 to 4 tokens,”

9.

In the sentence “...that have no bearing on the mood of the statement."

pleas replace 'mood' with 'meaning'.

10.

Section 3.2

“Encoding is highly required...”

please rephrase as

“Encoding is extensively required...”

11.

Section 5.Conclusion

Concerning the sentence “However, due to the magnitude of the data, the model's performance requires improvement.”

Please rephrase as

“Potentially, the model's performance could be improved by training on more data.”

12.

“Additional information from the holy books of Amharic...”

This is not quite clear, remember that the readers probably know little about this. Please clarify.

---

## [Author Response · Author response to Decision Letter 2]

20 Oct 2023

We corrected the document as per your comments and suggestions.

---

## [Editor Report · Decision Letter 3]

21 Nov 2023

Deep Learning-Based Idiomatic Expression Recognition for the Amharic Language

PONE-D-23-14519R3

Dear Dr. Endalie,

We’re pleased to inform you that your manuscript has been judged scientifically suitable for publication and will be formally accepted for publication once it meets all outstanding technical requirements.

Kind regards,

Michael Flor

Academic Editor

PLOS ONE

---

## [Editor Report · Acceptance letter]

2 Dec 2023

PONE-D-23-14519R3 

Deep Learning-Based Idiomatic Expression Recognition for the Amharic Language 

Dear Dr. Endalie:

I'm pleased to inform you that your manuscript has been deemed suitable for publication in PLOS ONE. Congratulations! Your manuscript is now with our production department. 

Kind regards, 

on behalf of

Dr. Michael Flor 

Academic Editor

PLOS ONE